# The Impact of Common Acne on the Well-Being of Young People Aged 15–35 Years and the Influence of Nutrition Knowledge and Diet on Acne Development

**DOI:** 10.3390/nu14245293

**Published:** 2022-12-13

**Authors:** Malgorzata Kostecka, Julianna Kostecka, Ola Szwed-Gułaga, Izabella Jackowska, Joanna Kostecka-Jarecka

**Affiliations:** 1Faculty of Food Science and Biotechnology, University of Life Sciences, Akademicka 15, 20-950 Lublin, Poland; 2Faculty of Medicine, Medical University of Lublin, Chodźki 19, 20-093 Lublin, Poland; 3Provincial Hospital Clinic, Dermatological Outpatient Clinic, Artwińskiego 1, 25-736 Kielce, Poland; 4Independent Public Healthcare Center in Łęczna, Krasnystawska 52, 21-010 Łęczna, Poland

**Keywords:** acne, well-being, HADS scores, anxiety, dietary factors, elimination diet

## Abstract

Acne is a disorder of sebaceous glands, and it most commonly develops on the face. The role of the diet in triggering and treating acne is controversial and has been widely debated in the literature. A knowledge of the environmental factors that contribute to acne could improve the patient’s physical and emotional well-being, increase the efficacy of treatment, and minimize the risk of anxiety and depressive disorders. The aim of this study was to assess the impact of acne on the daily lives and well-being of people aged 15–30 years, to analyze young people’s knowledge about the influence of various foods and other dietary factors on the prevalence, severity, and treatment of acne, as well as their adherence to an anti-acne diet. The study was conducted between April 2021 and May 2022. A total of 1329 respondents, including 963 women and 366 men, participated in the study. In 99% of men and women respondents, acne breakouts were typically located on the face. An analysis of HADS scores revealed moderate anxiety in 57% of women (F) and 22.5% of men (M) respondents. Acne breakouts located on the face were problematic for 81% of the study population (regardless of sex). More than ¾ of women and 2/3 of men claimed that acne made them feel less attractive. The impact of acne on the participants’ emotional well-being and social life differed between genders. Women experienced psychological discomfort more frequently than men (*p* = 0.0023). More than 50% of the participants eliminated acne-triggering foods from their diets, and 2/3 of these respondents observed a marked improvement or disappearance of skin breakouts as a result. A significantly higher number of respondents with severe/moderate anxiety were convinced that acne breakouts were affected by diet (OR 1.56; 95% CI 1.23–1.87, *p* < 0.001) and foods with a high glycemic index (OR 1.56; 95% CI 1.23–1.94, *p* < 0.001). Acne affects the patients’ emotional well-being. It can act as a barrier to social interactions and lead to mood disorders of varied severity. Persons with moderate/severe mood disorders associated with anxiety significantly more often recognized the role of dietary factors in acne aggravation, and the severity of mood disorders was directly correlated with more frequent consumption of sweets, sweetened beverages, and foods with a high glycemic index.

## 1. Introduction

The common acne (acne vulgaris) is one of the most widespread skin conditions in Europe and the United States, and the main reason for visiting a dermatologist [1]. Acne is a disorder of sebaceous glands, and it most commonly develops on the face [2]. The disease affects mostly young people between the ages of 12 and 25 (85% of the cases) [3]. Its prevalence is similar in both sexes, but severe acne is more frequently reported in men [4,5]. Most patients are diagnosed with mild acne, but severe acne is noted in 15% of the affected population, and it constitutes not only a medical problem, but also a social and a psychological issue [6] that can give rise to emotional problems, low self-esteem and, in the most severe cases, depression [7].

The growing interest in the human microbiome has spurred research into the role of microorganisms in the pathogenesis of skin diseases, including acne. *Cutibacterium acnes* is regarded as one of the key contributing factors, but its role remains unclear because this bacterium begins to colonize the sebaceous glands of healthy subjects in puberty. *Cutibacterium acnes* accounts for up to 90% of skin microbiota in areas with the highest concentration of sebacous glands, including the head, face, chest, and back [8]. This bacterium is rarely identified in childhood, and its counts increase gradually between puberty and adulthood, and decrease past the age of 50. *Cutibacterium acnes* helps maintain a low skin pH by releasing free fatty acids, and it prevents other pathogens (i.e., *Staphylococcus aureus* and *Streptococcus*) from colonizing the skin [9].

There is considerable scientific evidence to suggest that the gut microbiome affects the host’s overall health and physiology, and that the gut microbiome is less diverse in people with acne than in healthy subjects [10,11]. Gut microbiota probably contribute to acne by interacting with the mTOR signalling pathway [12,13], and bacterial metabolites may control cell growth, lipid metabolism, and other metabolic functions via the mTOR pathway [14]. Gut dysbiosis and a disrupted intestinal barrier can lead to a positive feedback loop which exacerbates inflammation [15,16]. 

In a study by Deng et al. [11], acne patients were characterized by a less diverse gut microbiome and a higher *Bacteroidetes* to *Firmicutes* ratio, which is typical of people consuming the Western diet [17]. *Actinobacteria* counts were found to be lower, whereas *Proteobacteria* counts were higher in persons with moderate to severe acne [18]. A recent clinical trial revealed that healthy subjects had a *Firmicutes* to *Bacteroidetes* ratio of 1, whereas acne patients had a ratio of 0.8 [19]. In addition, Yan et al. [19] reported a decrease in *Lactobacillus*, *Bifidobacterium*, *Butyricicoccus*, *Coprobacillus*, and *Allobaculum* counts in people with acne relative to the control group, which sheds a new light on the association between acne and changes in the gut microbiome. *Lactobacillus* and *Bifidobacterium* are ubiquitous probiotic bacteria that strengthen the gut barrier by decreasing intestinal permeability and enhancing epithelial immunity [20]. In contrast, another study revealed higher *Firmicutes* counts, a lower *Bacteroidetes* to *Firmicutes* ratio, and no differences in *Actinobacteria* and *Proteobacteria* counts in acne patients relative to healthy controls [21].

Acne formation can be influenced by environmental factors, including diet. Diet composition, the number of meals, and meal preparation methods can affect skin condition in multiple ways by modifying a person’s nutritional status, lipid and carbohydrate metabolism, hormone levels, and gut microbiome, and inducing specific immune responses [22]. The role of the diet in triggering and treating acne is controversial and has been widely debated in the literature, but the association between diet and acne is no longer disregarded. There is scientific evidence to indicate that acne is strongly linked with diet [23]. The Western diet appears to play a particularly important role in the pathogenesis of acne [24]. The Western diet is characterized by a high energy density and a high intake of meat, milk, dairy products, foods with a high glycemic index [25], foods rich in saturated fatty acids, and chocolate [11]. The Western diet is most popular in highly developed countries, where the prevalence of acne is also higher [26]. This skin condition does not affect populations living in less developed regions of the world and adhering to the paleolithic diet. Acne has never been reported on Kitava island in Papua New Guinea, Ache hunter-gatherers of Paraguay, Inuit people, or youths in rural Brazil [24]. A study of rural populations in Kenya and Zambia, and the Bantu people in southern Africa revealed that acne was less prevalent in the older generations than in the offspring who emigrated to the USA and adopted a Western diet, which could confirm the hypothesis that the Western diet contributes to acne [27]. 

A knowledge of the environmental factors that contribute to acne could improve the patient’s physical and emotional well-being, increase the efficacy of treatment, and minimize the risk of anxiety and depressive disorders. It appears that healthcare professionals should provide patients with reliable information about the ways in which the right diet can increase treatment efficacy and reduce the risk of acne relapse after treatment [28].

The aim of this study was to assess the impact of acne on the daily lives and well-being of people aged 15–30 years, to determine the prevalence of anxiety and depression in youths affected by acne, to analyze young people’s knowledge about the influence of various foods and other dietary factors on the prevalence, severity, and treatment of acne, as well as their adherence to an anti-acne diet.

## 2. Materials and Methods

### 2.1. Study Design and Participants

The study was conducted between April 2021 and May 2022 with the use of an online questionnaire (948 respondents) and a paper questionnaire that was completed by the patients of dermatology clinics in Lublin, Kielce, and Rzeszów in south-eastern Poland (381 respondents). The patients of dermatology clinics were invited to participate in the study and complete the questionnaire by surveyors and medical personnel. The questionnaire was addressed to men and women affected by acne. The inclusion criteria were age below 35 years, present or previous diagnosis of acne, and the absence of other active skin conditions such as psoriasis, contagious impetigo, urticaria, or atopic dermatitis. The purpose of the study and the target group were described on social media, and invitations were forwarded to persons who met the inclusion criteria. The link to the questionnaire was posted on Facebook and Instagram groups dedicated to acne and acne skin care. The paper questionnaire was completed independently by the patients of dermatology clinics without the researcher’s involvement.

### 2.2. Data Collection

The first part of the questionnaire contained 39 closed-ended single or multiple choice questions concerning the frequency of visits to a dermatologist or a cosmetologist, the respondents’ knowledge about dietary and lifestyle factors that can alleviate or aggravate acne lesions, the respondents’ diet, and the frequency of consumption of various food products (food frequency questionnaire—FFQ): [] never, [] 1–3 times per month, [] once a week, [] several times per week, [] daily, [] several times per day. The second part of the questionnaire focused on the impact of acne on the respondents’ physical and psychological well-being, and it contained the hospital anxiety and depression scale (HADS) self-assessment questionnaire which consists of eight questions for anxiety and eight questions for depression (a total of sixteen questions). The HASD had been initially designed for hospital patients with somatic disorders, but it is presently also recommended for outpatients. Within each subscale, a score of 0–7 points is considered normal, 8–10 points—as borderline abnormal, 11–14 points—as moderately abnormal, and 15–21 points—as abnormal [29,30,31]. The participants’ lifestyle and knowledge about acne were correlated with their HADS scores. The self-assessment of acne severity was based on the acne global severity scale, where 0 points denote normal, clear skin with no evidence of acne; 1 point denotes rare, non-inflammatory lesions and non-inflamed papules; 2 points denote some non-inflammatory lesions with few inflammatory lesions (papules/pustules only, no nodulocystic lesions); 3 points denote mostly non-inflammatory lesions, with multiple inflammatory lesions, including several to many comedones and papules/pustules, and/or a small number of nodulocystic lesions; 4 points denote more apparent inflammatory lesions, with many comedones and papules/pustules, and/or a small number of nodulocystic lesions; 5 points denote highly inflammatory lesions with a variable number of comedones, many papules/pustules, and many nodulocystic lesions. A score of 1 and 2 points is indicative of mild acne, 3 and 4 points—moderate acne, and 5 points—severe acne. 

### 2.3. Data Analysis

Categorical variables were presented as sample percentages (%). Before the statistical analysis, data were checked for normal distribution by the Kolmogorov–Smirnov test. The relationships between HADS scores, sex, place of residence, and the results of the questionnaire were analyzed with the use of Student’s *t*-test and ANOVA. A qualitative analysis of the relationships between questionnaire results and the participants’ sex and place of residence was performed using the χ^2^ test and Fisher’s exact test. The odds ratios (ORs) and 95% confidence intervals (95% CIs) were calculated. The reference categories (OR = 1.00) included the female sex, urban residents, university education, and the absence of mood disorders or mild anxiety/depressive disorders. The ORs were adjusted for diet composition and lifestyle factors that can aggravate or alleviate acne symptoms. Confounders were identified and categorized (place of residence and education). The significance of ORs was assessed by Wald’s statistics. The results of all tests were regarded as statistically significant at *p* < 0.05. Data were processed in the Statistica program (version 13.1 PL; StatSoft Inc., Tulsa, OK, USA; StatSoft, Krakow, Poland).

## 3. Results

A total of 1329 respondents participated in the study (Table 1). Most respondents, both men and women, had first experienced acne during puberty at the mean age of 15.3 years (14.9 for women, 15.5 for men, *p* > 0.05). Acne lesions on the chest/cleavage were more prevalent in women than men (41% of women vs. 11% of men, *p* < 0.05), whereas acne symptoms on the shoulders and the back were more prevalent in men than in women (52% in men vs. 17% in women, *p* < 0.001).

### 3.1. Anxiety and Depression in Patients with Acne

An analysis of HADS scores revealed moderate anxiety in 57% of women (F) and 22.5% of men (M) respondents, and moderate depression in 24% of women and 22% of men participants. Severe anxiety was reported by only 5% of women and 3.5% of men, and severe depression by 2% of women and 3% of men. Anxiety levels were significantly (*p* < 0.001) higher in women (11.43) than in men (8.86).

Acne breakouts located on the face were problematic for 81% of the study population (regardless of sex). More than ¾ of women and 2/3 of men claimed that acne made them feel less attractive, which indicates that this skin condition significantly compromises young people’s well-being. The impact of acne on the participants’ emotional well-being and social life differed between males and females. Women experienced psychological discomfort more frequently than men (*p* = 0.0023) and used make-up to conceal lesions, whereas men were more likely to experience physical discomfort related to physical activity (*p* = 0.007). In comparison with men, women more frequently visited dermatologists (62.3% F vs. 27.4% M, *p* < 0.001) and cosmetologists to improve the condition (48.2% F vs. 9.56% M, *p* < 0.001) and the appearance of their skin (58% F vs. 11.5% M, *p* < 0.001). Painful breakouts were reported by ¼ of the respondents, more frequently in women (*p* = 0.025), respondents experiencing moderate anxiety (*p* = 0.003), and men experiencing severe depression (*p* = 0.001). Nearly 2/3 of women and 20% of men found it difficult to share their emotions with their closest family members and friends (*p* < 0.001). Around 70% of women and 67% of the men experienced anger due to acne’s negative effects on their daily lives and mental or physical well-being (*p* = 0.314).

An analysis of the correlations between questionnaire responses and HADS scores (Table 2) revealed higher levels of anxiety in women, and it demonstrated that feelings of unattractiveness and embarrassment, and relationship problems were significantly correlated with severe anxiety (*p* = 0.0016). In males and females, physical discomfort was not linked with higher levels of depression (*p* = 0.0307) or anxiety (*p* = 0.078). The place of residence and the type of acne lesions were not correlated with increased risk of anxiety or depressive disorders; only in rural residents, severe mood disorders were associated with anxiety and not depression (*p* = 0.0011). Age was not correlated with anxiety (*p* = 0.713) or depression (*p* = 0.257) levels, probably because the study population was largely homogeneous in terms of age.

### 3.2. Knowledge about Foods That Affect Acne

Recent research has shown that diet and nutrition can alleviate or aggravate acne breakouts. According to the surveyed women, acne was aggravated mainly by fast foods (56%), chocolate (56%), spices (46%), salt (35.6%), and milk (38.7%). In turn, men listed spicy/salty snacks (72.9%), spices (67%), protein shakes (30%), and milk (21.6%) as the main acne triggers.

More than 50% of the participants eliminated acne-triggering foods from their diets, and 2/3 of these respondents observed a marked improvement or disappearance of skin breakouts as a result. Acne severity influenced the respondents’ decisions to switch to an elimination diet (*r* = 0.79; *p* < 0.001), and the duration of the elimination diet was associated with an improvement in skin condition (*r* = 0.81; *p* < 0.001). Adherence to the elimination diet was also associated with an improvement in psychological well-being (*r* = 0.76; *p* < 0.001). A significantly higher number of respondents with anxiety were convinced that acne breakouts were affected by diet (OR 1.56; 95% CI 1.23–1.87, *p* < 0.001) and foods with a high glycemic index (OR 1.56; 95% CI 1.23–1.94, *p* < 0.001) (Table 3). The place of residence significantly influenced the respondents’ knowledge about the role of nutrition in acne development and alleviation. Rural residents were significantly least likely to recognize the acne-triggering potential of certain foods (Table 4). 

A significantly higher number of respondents with moderate/severe depression (OR 1.27; 95% CI 1.11–1.46, *p* < 0.01) and men (OR 1.25; 95% CI 1.09–1.41, *p* < 0.01) were of the opinion that dairy products, in particular fermented milks (yogurt, kefir), alleviate acne breakouts. The above groups also recognized the positive role of probiotic bacteria in these food products. 

### 3.3. Diet and Dietary Habits of Persons with Acne

The frequency of consumption of basic food groups, with a particular emphasis on foods that can aggravate or alleviate acne symptoms, was evaluated based on the participants’ responses. 

In the surveyed group, more than 3/4 of the respondents consumed milk and dairy products. Most participants consumed milk and dairy products 3–4 times per week (48%), significantly more often women (*p* = 0.012) and respondents with primary education (*p* = 0.027). Plain yogurt, whole milk, lactose-free milk, and cottage cheese were the most frequently indicated. 

Foods with a high glycemic index were frequently consumed by the respondents. More than half of the study population, regardless of sex or education, consumed these products several times a week, and only 5% eliminated these foods completely (*p* = 0.007) because they aggravated skin breakouts. Acne flare-ups and the type of lesions had no effect on the consumption of sweets, and a significantly higher number of persons with moderate/severe anxiety ate sweets several times a day/once a day (*p* = 0.006).

More than 50% of the respondents ate fish several times a week, and fish was significantly more often consumed by women (*p* = 0.01), persons with university education (*p* = 0.023), and respondents with mild depression/anxiety. Only 5% of women and 8% of men ate fish several times a week, which is the currently recommended fish intake. 

Acne can be aggravated by fast foods, and 2/3 of the respondents ate these products several times a month. Men consumed fast foods several times a week, significantly more often than women (*p* = 0.04) although all participants were aware that fast foods contributed to skin problems.

An analysis of questionnaire results revealed that spices (mustard, red pepper, black pepper, vinegar) were consumed several times a month by most respondents, regardless of their place of residence, education, or severity of mood disorders. Most participants were aware that spices can aggravate acne, but only 7.2% of women completely eliminated these products from their diets. Salt intake was limited by only 15% of the respondents, more often by women (*p* = 0.001) and persons with university education (*p* = 0.034).

Supplements containing the probiotic strains of *Lactobacillus* and *Bifidobacterium* were used by 40% of the respondents. The most popular supplements were a mixture of several species of probiotic bacteria.

## 4. Discussion

The extent to which dietary factors contribute to acne development, progression, and treatment has long stirred controversy. Evidence-based dietary recommendations for persons with acne are scarce in the literature and dermatology textbooks. Far fewer studies have investigated the association between acne and diet than the efficacy of pharmacological acne treatments, probably due to the popular belief that foods and diet have a minor impact on acne and its treatment [32]; confounders such as depression, smoking, and alcohol consumption are also disregarded [33]. 

### 4.1. Factors That Aggravate Acne

Numerous studies have demonstrated that the Western diet characterized by increased intake of dairy products and foods with a high glycemic index affects the concentrations of hormones implicated in acne pathogenesis. Diets with a high glycemic index (GI > 55) were associated with poor glycemic control, elevated levels of postprandial insulin and insulin-like growth factor 1 (IGF-1), whereas diets with a low glycemic index were found to decrease fasting IGF-1 levels [34]. Smith et al. demonstrated that a diet with a low glycemic index (GI < 50) decreased acne severity and the number of skin breakouts in teenagers with mild and moderate acne during a 10-week dietary intervention [35]. In the present study, the respondents were aware that foods with a high glycemic index can exert a negative effect on skin condition and aggravate acne, but very few participants limited their intake of such products. Chocolate was not indicated as a potential acne trigger by the respondents. However, a recent randomized trial involving a small group of 14 young males revealed that chocolate can aggravate acne symptoms [36]. 

### 4.2. Factors That Alleviate Acne

The role of milk in acne formation remains ambiguous. In the current study, the respondents were of the opinion that milk and dairy products had a beneficial influence on acne lesions, and they indicated yogurt and whole milk as foods that alleviated skin inflammation. In contrast, La Rosa et al. did not observe a significant relationship between total dairy intake and the severity of acne breakouts, but they found that persons with acne consumed significantly more skim and low-fat milk [37]. Similar results were reported by Grossi et al. who noted no association between acne severity and consumption of cheese and yogurt [38]. Meta-analyses produced inconclusive results, but studies investigating the link between acne and dairy intake demonstrated that full-fat dairy products and whole milk had lower odds ratios, whereas low-fat/skim milk had higher odds ratios for acne [39]. 

Recent research has shown that dysbiosis of the gut microbiota can contribute to acne development. Deng et al. (2018) reported that gut microbiota were less diverse in persons with acne than in healthy individuals. In their study, the counts of bacteria belonging to the genera *Firmicutes, Clostridium, Lachnospiraceae,* and *Ruminococcaceae* were significantly lower, whereas *Bacteroidetes* counts were significantly higher in persons with acne than in the control group. A higher ratio of *Bacteroidetes* to *Firmicutes* bacteria, observed in acne sufferers, is also typical of persons consuming the Western diet [11,40]. Some researchers evaluated the efficacy of probiotics in acne treatment. In the present study, 40% of the respondents took dietary probiotic supplements to alleviate disease symptoms. Dysbiosis of the gut microbiome in acne patients was confirmed by Yan et al. [19]. In a study by Clark et al., patients receiving antibiotics as well as *Lactobacillus acidophilus* and *Bifidobacterium bifidum* probiotics reported better treatment outcomes and fewer side effects than control group patients who were not administered probiotics [41]. In the present study, the respondents recognized the positive impact of omega-3 fatty acids on acne. According to the surveyed population, fish oil or dietary supplements containing omega-3 fatty acids were more helpful in alleviating acne than saltwater fish or vegetable sources of these acids. In a 12-week trial conducted in the US, dietary supplements containing fish oil alleviated acne in 61% of the patients, but aggravated skin inflammations in 1/3 of the participants [42]. In a Korean study, the prevalence of inflammatory skin changes decreased considerably in subjects receiving fish oil and borage oil supplements [43]. In turn, the number of blackheads, whiteheads, red spots, and cysts as well as excess sebum production decreased in American teenagers who consumed high amounts of fish and seafood [44]. However, these studies do not provide unambiguous answers as to whether omega-3 fatty acids, regardless of acne severity or the applied treatment model, alleviate skin inflammations and acne breakouts, and they do not indicate the optimal dose, source, and duration of supplementation [45].

### 4.3. Stress and Depression in the Development and Aggravation of Acne

Stress is a potential trigger of acne vulgaris. In response to stress, peripheral nerves release substance P which leads to the proliferation and differentiation of sebaceous glands and increases lipid synthesis in sebocytes. In the current study, most of the women were characterized by moderate anxiety/stress levels, and men had mild anxiety/stress levels. In an Indonesian study, most acne patients had low (56.7%) and moderate (40%) scores on the stress scale [46]. Moderate stress scores were noted by Rokowska-Waluch et al. in the Polish population [47] and by Ratnasari et al. in the Balinese population [48]. Acne breakouts and scars on the face influence the patients’ emotional state and anxiety levels. In this study, feelings of unattractiveness and embarrassment were closely associated with higher anxiety levels. Similar results were reported by Lee et al. [49] who investigated whether acne patients demonstrated greater psychological bias when assessing the attractiveness of faces with acne lesions, and whether they devoted more selective attention to acne breakouts than acne-free individuals in the control group. They found that acne patients had a stronger attentional bias for acne lesions than control individuals. The psychological and emotional implications of acne should be taken into consideration by dermatologists. Similar conclusions were formulated by Timms who observed that in young people, moderate acne can act a potential barrier to social relationships due to social anxiety and discriminatory attitudes of acne-free peers [50]. 

Acne is associated with lower perceptions of physical attractiveness and lower self-esteem [36,37], and it can contribute to internalized stigma (anxiety and depression) [42]. Three large-population studies [51,52,53] revealed that the prevalence of clinical depression was higher in acne patients. However, most researchers have reported contradictory results, and they did not examine a full range of factors (age, trial conditions, treatment, geographical region) due to differences in methodology [54]. In the present study, approximately ¼ of the patients reported symptoms of moderate depression, regardless of age or sex. Samuels et al. reported surprising results which suggest that depression and anxiety are more prevalent among adults than teenagers with acne [7]. Similar observations had been previously made by Yang et al. (2014) and Murray et al. (2005) who found that adult acne patients complained of aggravated stress caused by social alienation and the popular belief that acne affects only teenagers [55,56].

### 4.4. Strengths and Limitations

A large group of respondents, including a high percentage of men participants, was a strength of this study. The HADS was used to determine the prevalence of mood disorders, including anxiety and depression, in acne sufferers and a questionnaire was completed by the respondents to evaluate how they coped with acne; therefore, the associations between the analyzed factors could be analyzed. This is also the first large-population study in Poland to assess acne sufferers’ knowledge about dietary factors that influence acne, and the results can be used by physicians and nutritionists to develop new treatment strategies and educate patients about acne. 

However, the study also has several weaknesses. The effects of the type, duration, and efficacy of the administered pharmacological treatment on the patients’ emotional well-being were not analyzed, and potential food–drug interactions were not investigated. Another limitation of the study was the absence of information about ongoing treatment with oral or topical antibiotics or oral or topical retinoids. The role of individual dietary components that can potentially modify drug metabolism, thus affecting treatment, was not studied, either.

## 5. Conclusions

Acne affects the patients’ emotional well-being. It can act as a barrier to social interactions and lead to mood disorders of varied severity.

Acne lesions on the face posed a problem for most respondents and compromised self-perceived attractiveness.

Persons with moderate/severe mood disorders associated with anxiety significantly more often recognized the role of dietary factors in acne aggravation, and the severity of mood disorders was directly correlated with more frequent consumption of sweets, sweetened beverages, and foods with a high glycemic index.

Respondents who adhered to an elimination diet for more than 12 months during acne treatment more frequently reported a complete elimination of skin lesions; however, more than 50% of the surveyed subjects found it difficult to adhere to an elimination diet on a regular basis. 

Milk and dairy consumption was high in the study population, in particular in men and persons with moderate/severe depression.

According to nearly 50% of the respondents, probiotic supplements containing *Lactobacillus* or *Bifidobacterium* strains can alleviate acne symptoms.

Healthcare professionals should provide patients with reliable information about the ways in which the right diet can increase treatment efficacy and reduce the risk of acne relapse after treatment. Nutritional consultations for acne sufferers could improve their knowledge about dietary components that alleviate or aggregate acne, and they could assist patients in planning and adhering to an anti-acne diet. 

## Figures and Tables

**Table 1 nutrients-14-05293-t001:** Characteristics of the study population.

Respondents			*p*
Sex *n* [%]	Women	963 [72.4%]	<0.01
	Men	366 [27.6%]	
Time when acne symptoms first appeared	In the previous 5 years	731 [55%]	<0.01
	In the previous 2–3 years	518 [39%]	
	In the previous year	80 [6%]	
Closest family members were/are also affected by acne	Yes	877 [66%]	<0.01
	No	452 [34%]	
Location of lesions [multiple answers possible]	Face	1315 [99%]	<0.01
	Chest/cleavage	463 [34.8%]	
	Shoulders and back	354 [26.6%]	
Type of lesions [multiple answers possible]	Blackheads	585 [44%]	<0.01
	Whiteheads	492 [37%]	
	Red spots	106 [8%]	
	Cysts	146 [11%]	
Acne severity	Severe	186 [14%]	<0.01
	Moderate	1023 [77%]	
	Mild	120 [9%]	

**Table 2 nutrients-14-05293-t002:** Relationships between the respondents’ characteristics, the impact of acne on the respondents’ well-being, and HADS scores.

		Anxiety		*p*	Depression		*p*
		Mean	SD		Mean	SD	
Sex	F	11.43	2.34	<0.001	8.97	2.21	ns
	M	8.86	2.27		8.91	3.04	
Acne severity	Severe	9.78	2.71	0.001	8.97	2.56	ns
	Moderate	9.21	2.34		8.75	2.81	
	Mild	8.76	2.31		8.91	2.48	
Have you ever visited a dermatologist?	Yes	9.07	2.31	ns	11.46	2.56	0.0023
	No	9.24	2.76		9.17	2.44	
Have you ever visited a cosmetologist?	Yes	9.53	3.15	ns	10.11	2.51	ns
	No	9.41	2.89		10.43	2.49	
Does acne make you feel angry?	Yes	10.56	2.50	0.001	11.74	3.57	0.0017
	No	8.41	2.34		8.81	2.46	
Does acne cause psychological discomfort (embarrassment, frustration)?	Yes	9.73	3.06	0.0024	9.11	2.71	ns
	No	8.41	2.51		8.87	2.46	
Does acne cause physical discomfort (pain, itching, burning)?	Yes	9.01	3.41	ns	8.71	2.79	ns
	No	8.86	2.97		8.54	2.47	
Do you think that other people pay attention to skin blemishes?	Yes	11.41	3.36	0.0011	10.11	3.11	0.0013
	No	9.17	2.17		8.44	2.57	
Do visible skin blemishes negatively impact personal relationships?	Yes	10.56	3.11	0.0073	9.17	2.71	ns
	No	9.14	2.47		9.03	2.59	
Can visible skin blemishes lead to rejection and social isolation?	Yes	10.01	3.13	0.024	8.97	2.91	ns
	No	9.23	2.70		9.05	3.16	
Can visible skin blemishes negatively affect employment opportunities?	Yes	9.24	2.70	ns	8.47	2.56	ns
	No	9.31	2.56		8.70	2.49	

ns—not statistically significant.

**Table 3 nutrients-14-05293-t003:** Respondent opinions on diet and lifestyle factors that can aggravate acne.

	Men (Ref. Women)	Rural Residents (Ref. Urban Residents)	Respondents with Primary Education (Ref. Respondents with University Education)	Respondents with Moderate and Severe Anxiety (Ref. Respondents without Anxiety or with Mild Anxiety)	Respondents with Moderate and Severe Depression (Ref. Respondents without Depression or with Mild Depression)
Certain foods can aggravate acne	1.12 (0.89–1.27)	0.84 * (0.71–1.07)	0.79 * (0.64–1.09)	1.56 ** (1.23–1.87)	1.04 (0.91–1.17)
Omega-6 fatty acids can aggravate acne	0.82 * (0.73–0.97)	0.79 * (0.64–0.91)	1.09 (0.94–1.17)	1.12 (0.94–1.21)	1.03 (0.88–1.12)
Gluten-containing products can trigger acne breakouts	1.17 (0.94–1.26)	0.91 (0.78–1.06)	0.94 (0.87–1.17)	1.21 * (1.03–1.37)	1.06 (0.91–1.19)
Foods with a high glycemic index can aggravate acne	0.76 * (0.61–0.94)	0.51 ** (0.44–0.81)	0.91 (0.74–1.19)	1.56 ** (1.23–1.94)	1.30 * (1.17–1.61)
Nuts can aggravate acne	1.07 (0.87–1.20)	0.87 * (0.70–1.04)	0.71 * (0.63–0.94)	1.07 (0.88–1.24)	1.27 * (1.05–1.41)
Smoking can aggravate acne	1.24 * (1.01–1.46)	1.11 (0.94–1.260	0.89 * (0.76–1.14)	1.09 (0.94–1.27)	1.17 (0.90–1.31)
Alcohol can aggravate acne	0.74 * (0.61–0.97)	0.79 * (0.59–1.07)	0.90 (0.84–1.11)	1.07 (0.89–1.24)	1.11 (0.91–1.27)

Two confounders were considered in the study: place of residence (respondents living in towns and cities with a population higher than 20,000 were classified as urban residents, and the remaining respondents—as rural residents) and education (respondents who passed the matriculation exam were placed in the secondary education group; respondents with a university diploma were placed in the university education group; respondents who did not pass the matriculation exam or were still in school were placed in the primary education group). *p* < 0.05 *; *p* < 0.01 **.

**Table 4 nutrients-14-05293-t004:** Respondent opinions on diet and lifestyle factors that can alleviate acne.

	Men (Ref. Women)	Rural Residents (Ref. Urban Residents)	Respondents with Primary Education (Ref. Respondents with University Education)	Respondents with Moderate and Severe Anxiety (Ref. Respondents without Anxiety or with Mild Anxiety)	Respondents with Moderate and Severe Depression (Ref. Respondents without Depression or with Mild Depression)
Certain foods can alleviate acne	1.15 (0.94–1.29)	0.91 (0.84–1.19)	0.84 * (0.71–0.99)	1.29 * (0.84–1.41)	1.13 (0.88–1.30)
Omega-3 fatty acids can alleviate acne	0.92 (0.81–1.09)	0.94 (0.81–1.19)	0.59 ** (0.41–0.97)	1.11 (0.97–1.29)	1.27 * (1.03–1.39)
Products containing antioxidant vitamins (A, D, E) can alleviate acne	1.10 (0.94–1.21)	1.04 (0.91–1.17)	1.12 (0.88–1.27)	1.06 (0.89–1.13)	1.01 (0.91–1.17)
Zinc and selenium can alleviate acne	0.84 * (0.71–1.13)	0.70 * (0.63–0.91)	0.54 ** (0.37–0.73)	1.12 (0.94–1.23)	1.07 (0.90–1.19)
Dairy products can alleviate acne	1.25 * (0.89–1.41)	0.88 * (0.74–0.97)	1.14 (0.86–1.27)	1.46 * (1.17–1.61)	1.27 * (1.11–1.45)
Probiotics and probiotic foods can alleviate acne	1.04 (0.88–1.15)	1.06 (0.91–1.12)	0.74 * (0.61–0.93)	1.11 (0.92–1.20)	1.36 * (0.99–1.54)
Probiotics supplements can alleviate acne	1.08 (0.91–1.21)	1.03 (0.94–1.14)	0.65 * (0.57–0.81)	1.06 (0.95–1.15)	1.40 * (1.23–1.67)
*Lactobacillus bifidus* can alleviate acne	1.07 (1.01–1.12)	1.03 (0.89–1.11)	0.81 * (0.76–0.93)	1.32 * (1.17–1.43)	1.19 (1.03–1.27)

Three confounders were considered in the study: place of residence (respondents living in towns and cities with a population higher than 20,000 were classified as urban residents, and the remaining respondents—as rural residents), education (respondents who passed the matriculation exam were placed in the secondary education group; respondents with a university diploma were placed in the university education group; respondents who did not pass the matriculation exam or were still in school were placed in the primary education group), and severity of mood disorders (respondents without mood disorders and with mild mood disorders were placed in the first group, whereas respondents with moderate and severe mood disorders were placed in the second group). *p* < 0.05 *; *p* < 0.01 **.

## Data Availability

Due to ethical restrictions and participant confidentiality, data cannot be made publicly available.

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
