# Peer review of "The Impact of Common Acne on the Well-Being of Young People Aged 15–35 Years and the Influence of Nutrition Knowledge and Diet on Acne Development"

_nutrients, 2022, doi:10.3390/nu14245293_

Round 1

Reviewer 1 Report

This is an interesting study on the impact of diet and nutrition knowledge on acne in a young population of patients between 15 and 30.

A few questions arise:

1)  How many were acne patients from the acne clinics and filled out paper questionnaires vs online responders?

2)  I would recommend breaking the data down by those from the dermatology clinics with good clinical analysis and evaluation vs those with self- examination by responders online. 

3)  I am curious on the difference in response in those with more severe acne and with little to no acne as well as difference in depression and anxiety.  How was the severity assessed.

4)  I am unclear how data could be collected on how diet modifications could impact acne since the methods appear to describe a single questionnaire for diet and impact and separate questionnaire for depresssion / anxiety that are taken at a single point in time.  Please clarify.

5)  Were the online responders asked to verify their acne by lesion and distribution and count through photos or other mechanisms? How were actual numbers of lesions deternined?  How were these verified?

6)  Were data on any ongoing treatment with oral or topical antibiotics or oral or topical retinoinds being collected?

7) Did the questions also include questions on the actual diet of the responders or just nutrition knowledge (for both the online subset as well as the acne patient subset)?

8) How were potential online responders targeted or invited to participate?

9)  How were acne patients recruited for the study

10)  Was any validation performed for the depression and anxiety?  Or were these diagnoses based on answers to questions?  Or were these diagnoses based on self-assessment of these diagnoses?

11)  I am unclear on how adherence to an anti-acne diet was monitored.  Was this subjective based on self-reporting?

12)  How were flare-ups defined and assessed?

13)  I am unclear on how the severity of the mood disorders was assessed.

Author Response

We are grateful to the Reviewers for their insightful and detailed comments. The Reviewers' remarks were considered in the revision process, and we hope that the introduced changes have improved the quality of the manuscript.  In the following, we highlight (in green) comments and suggestions from Reviewer 1 and our effort to address these concerns.

How many were acne patients from the acne clinics and filled out paper questionnaires vs online responders?

The study involved 381 patients of dermatology clinics and 948 respondents who completed the online questionnaire.

 I would recommend breaking the data down by those from the dermatology clinics with good clinical analysis and evaluation vs those with self- examination by responders online. 

Thank you for this suggestion. The answers given by dermatology clinic patients and online respondents were not compared because in both cases, the questionnaires were completed independently without any assistance from surveyors, physicians, or other medical personnel.

I am curious on the difference in response in those with more severe acne and with little to no acne as well as difference in depression and anxiety.  How was the severity assessed.

The responses were assessed based on the severity of acne, based on the self-assessment made by the respondents

I am unclear how data could be collected on how diet modifications could impact acne since the methods appear to describe a single questionnaire for diet and impact and separate questionnaire for depression / anxiety that are taken at a single point in time.  Please clarify.

We agree that the questionnaire was insufficiently described. The questionnaire was described in greater detail in the revised manuscript. The questionnaire consisted of two parts. The respondents' nutrition knowledge and dietary habits were assessed in the first part. The second part contained the HADS self-assessment questionnaire to evaluate the impact of acne on the respondents' psychological well-being.

Were the online responders asked to verify their acne by lesion and distribution and count through photos or other mechanisms? How were actual numbers of lesions determined?  How were these verified?

Thank you for this suggestion. The self-assessment of acne severity was described in greater detail in the revised manuscript. Both clinic patients and online respondents assessed the severity of acne on the Acne Global Severity Scale.

Were data on any ongoing treatment with oral or topical antibiotics or oral or topical retinoids being collected?

No, ongoing treatment was not taken into consideration, but this is a valuable suggestion for further research. This problem was described as a limitation of the study.

Did the questions also include questions on the actual diet of the responders or just nutrition knowledge (for both the online subset as well as the acne patient subset)?

As indicated in the Materials and Methods section, the respondents were asked to indicate how frequently they consumed foods that can trigger or alleviate acne. The respondents did not keep food diaries, and they were only asked to indicate how frequently they consumed selected food products.

How were potential online responders targeted or invited to participate?

The relevant information was provided in the revised manuscript, thank you for pointing this out.

How were acne patients recruited for the study

Information about the study was posted in dermatology clinics. The surveyors and medical personnel encouraged patients to complete paper or online questionnaires. The relevant information was provided in the revised manuscript.

Was any validation performed for the depression and anxiety?  Or were these diagnoses based on answers to questions?  Or were these diagnoses based on self-assessment of these diagnoses?

Anxiety and depression scale was evaluated based on the respondents' HADS scores. A self-assessment method was applied.

 I am unclear on how adherence to an anti-acne diet was monitored.  Was this subjective based on self-reporting?

Thank you for raising this question.

Adherence to an anti-acne diet was not monitored. Only self-reported frequency of consumption of selected food products was evaluated, as mentioned above. The results were used to determine whether elimination of selected food products/adherence to the diet was difficult/was effective in alleviating acne symptoms/did not improve skin condition. The respondents were also asked to indicate the duration of the elimination diet. Long-term, regular contact with acne patients, preferably in a dermatology clinic, would be required to monitor the respondents' adherence to an anti-acne diet. Such an approach could be adopted in our next study.

How were flare-ups defined and assessed?

Flare-ups were defined as the worsening of skin condition and the appearance of acne symptoms. Acne symptoms were self-assessed on the

Acne Global Severity Scale.

I am unclear on how the severity of the mood disorders was assessed.

The severity of mood disorders was not assessed. The impact of acne on the respondents' psychological well-being was evaluated with the use of the validated HADS containing two subsets of questions (subscales) for anxiety and depression.

Reviewer 2 Report

The title is misleading. Microbiome is not examined in the project. This manuscript does not fit the microbiome special issue.

Abstract is extremely long. Is the word limit followed?

Men and women, male and female, use the same term

Microbiome

Need a reference for line 76.

Belong to result, line 117.

Divide 2.2 to study participants and data collection. Or add data collection part.

How many survey were done online, and how many were done in the clinic.  Are the characteristics of study participants different based on how the survey is completed?  

The results are not easy to follow. Was the frequency of food intake analysis only? Have a statistician to rewrite the data analysis part and recheck the analysis. What are the major outcomes of the research? How the criteria for confounding factors were defined? The results should be reorganized for clarity and conciseness. Try to avoid use correlation in this analysis, instead use difference.

84% of participants were 15-30 years old. Was the analysis focused on this group of participants?

“Study population”, not “studied population”

Lines 148 and 149 please rephrase.

What is the hypothesis for Table 1? What is the mean and P value for? Was the P value for the gender difference of anxiety by the acne severity?   Rephrase the title 1 to be more complete. If it is moderate anxiety or depression, please show in the table. Authors mixed moderate and severe together. 

Line 157, it is not correlate. Chi2 test dose not test for correlation. “not differ by sex”.

If line 157 to line 159 were not shown in Table 1, please use the other paragraph.

Table 1 should be used to show general characteristics of participants.

Line 215, this statement is too strong and is not justified.

If milk alleviates acne, should it be included in Section 4.2?

Dairy product alleviates or makes it more severe. The findings on dairy product are not consistently shown.

The authors may want to focus on the most important findings to present.

Author Response

We are grateful to the Reviewers for their insightful and detailed comments. The Reviewers' remarks were considered in the revision process, and we hope that the introduced changes have improved the quality of the manuscript.  In the following, we highlight (in violet) comments and suggestions from Reviewer 2 and our effort to address these concerns.

The title is misleading. Microbiome is not examined in the project. This manuscript does not fit the microbiome special issue.

Abstract is extremely long. Is the word limit followed?

we added a paragraph, as suggested by the Editor, regarding the role of the microbiome and probiotic bacteria in skin diseases, including acne

We agree with the Reviewer. The abstract was shortened.

Men and women, male and female, use the same term

The same terms were used consistently in the revised manuscript.

Need a reference for line 76.

The reference was provided.

Belong to result, line 117.

This line describes the scope of the study, therefore in our opinion it belongs to the Materials and Methods section.

Divide 2.2 to study participants and data collection. Or add data collection part.

Thank you for this suggestion. The participants and data collection were described in separate subsections.

How many survey were done online, and how many were done in the clinic.  Are the characteristics of study participants different based on how the survey is completed?  

The relevant information was provided in the revised manuscript. The characteristics of the study participants did not differ based on how the survey was conducted. Both clinic patients and online respondents completed the questionnaire independently to assess their well-being and dietary habits.

The results are not easy to follow. Was the frequency of food intake analysis only? Have a statistician to rewrite the data analysis part and recheck the analysis. What are the major outcomes of the research? How the criteria for confounding factors were defined? The results should be reorganized for clarity and conciseness. Try to avoid use correlation in this analysis, instead use difference.

Thank you for these suggestions. The Results section was shortened. The values of Pearson's r were provided. The severity of mood disorders was added as the third confounding factor (the respondents were divided into two groups based on the severity of mood disorders). Subsection 3.3 (Diet and dietary habits of persons with acne) was shortened, and Table 1 was modified. We hope that the results are now easier to follow.

84% of participants were 15-30 years old. Was the analysis focused on this group of participants?

“Study population”, not “studied population”

No, the study was conducted on the general population, but people aged 15-30 years were the largest group.

The relevant corrections were made.

Lines 148 and 149 please rephrase.

The sentence was rephrased.

What is the hypothesis for Table 1? What is the mean and P value for? Was the P value for the gender difference of anxiety by the acne severity?   Rephrase the title 1 to be more complete. If it is moderate anxiety or depression, please show in the table. Authors mixed moderate and severe together. 

Table 1 was thoroughly modified, and it now presents the characteristics of the study population.

Line 157, it is not correlate. Chi2 test dose not test for correlation. “not differ by sex”.

If line 157 to line 159 were not shown in Table 1, please use the other paragraph.

Thank you for this remark. The indicated lines were revised.

Table 1 should be used to show general characteristics of participants.

Thank you for this remark. Table 1 was thoroughly revised.

Line 215, this statement is too strong and is not justified.

The statement was modified.

If milk alleviates acne, should it be included in Section 4.2?

Dairy product alleviates or makes it more severe. The findings on dairy product are not consistently shown.

Thank you for this observation. Most respondents were of the opinion that milk and dairy products alleviated acne, but published findings are not conclusive. Subsections 4.1 and 4.2 were revised.

The authors may want to focus on the most important findings to present.

Subsection 3.3 (Diet and dietary habits of persons with acne) was shortened to present only the most important findings.